# A fairer way to compare researchers at any career stage and in any discipline using open-access citation data

**Corey J. A. Bradshaw**[1,2]*, **Justin M. Chalker**[3], **Stefani A. Crabtree**[4,5,6], **Bart A. Eijkelkamp**[7], **John A. Long**[7], **Justine R. Smith**[8], **Kate Trinajstic**[9], **Vera Weisbecker**[2,7]

**1** Global Ecology, College of Science and Engineering, Flinders University, Adelaide, South Australia, Australia, **2** ARC Centre of Excellence for Australian Biodiversity and Heritage, EpicAustralia.org, Adelaide, Australia, **3** Institute for Nanoscale Science and Technology, College of Science and Engineering, Flinders University, Adelaide, South Australia, Australia, **4** Department of Environment and Society, Utah State University, Logan, Utah, United States of America, **5** The Santa Fe Institute, Santa Fe, New Mexico, United States of America, **6** The Centre for Research and Interdisciplinarity, Paris, France, **7** College of Science and Engineering, Flinders University, Adelaide, South Australia, Australia, **8** Eye and Vision Health, Flinders University College of Medicine and Public Health, Flinders University, Adelaide, South Australia, Australia, **9** School of Molecular and Life Sciences, Curtin University, Bentley, Western Australia, Australia

* corey.bradshaw@flinders.edu.au

**Data Availability Statement:** Example code and data to calculate the index are available at: github.com/cjabradshaw/EpsilonIndex. A R Shiny application is also available at: cjabradshaw.

## Abstract

The pursuit of simple, yet fair, unbiased, and objective measures of researcher performance has occupied bibliometricians and the research community as a whole for decades. However, despite the diversity of available metrics, most are either complex to calculate or not readily applied in the most common assessment exercises (e.g., grant assessment, job applications). The ubiquity of metrics like the $h$-index ($h$ papers with at least $h$ citations) and its time-corrected variant, the $m$-quotient ($h$-index ÷ number of years publishing) therefore reflect the ease of use rather than their capacity to differentiate researchers fairly among disciplines, career stage, or gender. We address this problem here by defining an easily calculated index based on publicly available citation data (Google Scholar) that corrects for most biases and allows assessors to compare researchers at any stage of their career and from any discipline on the same scale. Our $\varepsilon'$-index violates fewer statistical assumptions relative to other metrics when comparing groups of researchers, and can be easily modified to remove inherent gender biases in citation data. We demonstrate the utility of the $\varepsilon'$-index using a sample of 480 researchers with Google Scholar profiles, stratified evenly into eight disciplines (archaeology, chemistry, ecology, evolution and development, geology, microbiology, ophthalmology, palaeontology), three career stages (early, mid-, late-career), and two genders. We advocate the use of the $\varepsilon'$-index whenever assessors must compare research performance among researchers of different backgrounds, but emphasize that no single index should be used exclusively to rank researcher capability.

shinyapps.io/epsilonIndex, with its relevant code and example dataset available at: github.com/cjabradshaw/EpsilonIndexShiny.

**Funding:** The authors received no specific funding for this work.

**Competing interests:** The authors declare no competing interests.

## Introduction

Deriving a fair, unbiased, and easily generated quantitative index serving as a reasonable first-pass metric for comparing the relative performance of academic researchers is—by the very complexity, diversity, and intangibility of research output across academic disciplines—impossible [1]. However, that unachievable aim has not discouraged bibliometricians and non-bibliometricians alike from developing scores of citation-based variants [2–4] in an attempt to do exactly that, from the better-known $h$-index [5, 6] ($h$ papers with at least $h$ citations), $m$-quotient [5, 6] ($h$-index ÷ number of years publishing), and $g$-index [7] (unique largest number such that the top $g$ papers decreasingly ordered by citations have least $g^2$ citations), to the scores of variants of these and other indices—e.g., $h^2$-index, $e$-index [8], $\chi$-index [9], $h_m$-index [10], $g_m$-index [11], etc. [3]. Each metric has its own biases and strengths [12–14], suggesting that several should be used simultaneously to assess citation performance. For example, the arguably most-popular $h$-index down-weights quality relative to quantity [15], ignores the majority of accumulated citations in the most highly cited papers [16], has markedly different distributions among disciplines [17], and tends to increase with experience [18]. As such, It has been argued that the $h$-index should not be considered for ranking a scientist's overall impact [19]. The $h$-index can even rise following the death of the researcher, because the $h$-index can never decline [2] and citations can continue to accumulate posthumously.

Despite their broad use for *inter alia* assessing candidates applying for academic positions, comparing the track records of researchers applying for grants, to applications for promotion [3, 20], single-value citation metrics are rarely meant to (nor should they) be definitive assessment tools [3]. Instead, their most valuable (and fairest) application is to provide a quick 'first pass' to rank a sample of researchers, followed by more detailed assessment of publication quality, experience, grant successes, mentorship, collegiality and all the other characteristics that make a researcher more or less competitive for rare positions and grant monies. But despite the many different metrics available and arguable improvements that have been proposed since 2005 when the $h$-index was first developed [5, 6], few are used regularly in these regards. This is because they are difficult to calculate without detailed data of a candidate's publication history, they are not readily available on open-access websites, and/or they tend to be highly correlated with the $h$-index anyway [21]. It is for these reasons that the admittedly flawed [19, 22, 23] $h$-index and its experienced-corrected variant, the $m$-quotient, are still the dominant ($h$-index much more so than the $m$-quotient) [2] metrics employed given that they are easily calculated [2, 24] and found for most researchers on open-access websites such as Google Scholar [25] (scholar.google.com). The lack of access and detailed understanding of the many other citation-based metrics mean that most of them go unused [3], and are essentially valueless for every-day applications of researcher assessment.

The specific weaknesses of the $h$-index or $m$-quotient make the comparison of researchers in different career stages, genders, and disciplines unfair because they are not normalized in any way. Furthermore, there is no quantitatively supported threshold above or below which assessors can easily ascertain minimum citation performance for particular applications—while assessors certainly use subjective 'rules of thumb', a more objective approach is preferable. For this reason, an ideal citation-based metric should only be considered as a relative index of performance, but relative to what, and to whom?

To address these issues and to provide assessors with an easy, rapid, yet objective *relative* index of citation performance for any group of researchers, we designed a new index we call the '$\varepsilon$-index' (the '$\varepsilon$' signifies the use of residuals, or deviance from a trend) that is simple to construct, can be standardized across disciplines, is meaningful only as a relative index for a particular sample of researchers, can be corrected for career breaks (see Methods), and

provides a sample-specific threshold above and below which assessors can determine whether individual performance is greater or less than that expected relative to the other researchers in the specific sample.

With the R code and online app we provide, an assessor need only acquire four separate items of information from Google Scholar (or if they have access, from other databases such as Scopus—scopus.com) to calculate a researcher's $\varepsilon$-index: (*i*) the number of citations acquired for the researcher's top-cited paper (i.e., the first entry in the Google Scholar profile), (*ii*) the *i*10-index (number of articles with at least 10 citations), (*iii*) the *h*-index, and (*iv*) the year in which the researcher's first peer-reviewed paper was published. While the last item requires sorting a researcher's outputs by year and scrolling to the earliest paper, this is not a time-consuming process. We demonstrate the performance of the $\varepsilon$-index using Google Scholar citation data we collected for 480 researchers in eight separate disciplines spread equally across genders and career stages to show how the $\varepsilon$-index performs relative to the *m*-quotient (the only other readily available, opportunity-corrected citation index available on Google Scholar) across disciplines, career stages, and genders. We also provide a simple method to scale the index across disciplines ($\varepsilon'$-index) to make researchers in different areas comparable despite variable citation trends within their respective areas.

## Materials and methods

### Researcher samples

Each co-author assembled an example set of researchers from within her/his field, which we broadly defined as *archaeology* (S.A.C.), *chemistry* (J.M.C.), *ecology* (C.J.A.B.), *evolution/development* (V.W.), *geology* (K.T.), *microbiology* (B.A.E.), *ophthalmology* (J.R.S.), and *palaeontology* (J.A.L.). Our basic assembly rules for each of these discipline samples were: (*i*) 20 researchers from each stage of career, defined here arbitrarily as *early career* (0–10 years since first peer-reviewed article published in a recognized scientific journal), *mid-career* (11–20 years since first publication), and *late career* (> 20 years since first publication); each discipline therefore had a total of 60 researchers, for a total sample of 8 × 60 = 480 researchers across all sampled disciplines. (*ii*) Each sample had to include an equal number of women and men from each career stage. (*iii*) Each researcher had to have a unique, publicly accessible Google Scholar profile with no obvious errors, inappropriate additions, obvious omissions, or duplications. The entire approach we present here assumes that each researcher's Google Scholar profile is accurate, up-to-date, and complete.

We did not impose any other rules for sample assembly, but encouraged each compiler to include only a few previous co-authors. Our goal was to have as much 'inside knowledge' as possible with respect to each discipline, but also to include a wide array of researchers who were predominantly independent of each of us. The composition of each sample is somewhat irrelevant for the purposes of our example dataset; we merely attempted gender and career-level balance to show the properties of the ranking system (i.e., we did not intend for sampling to be a definitive comment about the performance of particular researchers, nor did we mean for each sample to represent an entire discipline). Finally, we completely anonymized the sample data for publication.

### Citation data

Our overall aim was to provide a meaningful and objective method for ranking researchers by citation history without requiring extensive online researching or information that was not easily obtainable from a publicly available, online profile. We also wanted to avoid an index

that was overly influenced by outlier citations, while still keeping valuable performance information regarding high-citation outputs and total productivity (number of outputs).

For each researcher, the algorithm requires the following information collected from Google Scholar: (*i*) **i10-index** (the number of publications in the researcher's profile with at least 10 citations, which we denoted $i_{10}$); one condition is that a researcher must have $i_{10} \geq 1$ for the algorithm to function correctly; (*ii*) **h-index**—the researcher's Hirsch number [5]: the number of publications with at least as many citations, which we denoted *h*; (*iii*) the number of citations for the researcher's most highly cited paper (denoted **$c_m$**); and (*iv*) the year the researcher published her/his first peer-reviewed article in a recognized scientific journal (denoted **$Y_1$**). For the designation of $Y_1$, we excluded any reports, chapters, books, theses or other forms of publication that preceded the year of the first peer-reviewed article; however, we included citations from the former sources in the researcher's $i_{10}$, *h*, and $c_m$.

## Ranking algorithm

The algorithm first computes a power-law-like relationship between the vector of frequencies (as measured from Google Scholar): $i_{10}$, *h*, and 1, and the vector of their corresponding values: 10, *h*, and $c_m$, respectively. Thus, *h* is, by definition, both a frequency (*y*-axis) and value (*x*-axis). We then calculated a simple linear model of the form $y \sim \alpha + \beta x$, where

$$y = \log_e \begin{bmatrix} i_{10} \\ h \\ 1 \end{bmatrix} \text{ and } x = \log_e \begin{bmatrix} 10 \\ h \\ c_m \end{bmatrix}$$

(*y* is the citation frequency, and *x* is the citation value) for each researcher (S1 Fig). The corresponding $\hat{\alpha}$ and $\hat{\beta}$ for each relationship allowed us to calculate a standardized integral (area under the power-law relationship, $A_{rel}$) relative to the researcher in the sample with the highest $c_m$. Here, the sum of the predicted *y* derived from incrementing values of *x* (here in units of 0.05) using $\hat{\alpha}$ and $\hat{\beta}$ is divided by the product of $c_m$ and the number of incremental *x* values. This implies all areas were scaled to the maximum in the sample, but avoids the problem of truncating variances near a maximum of 1 had we used the maximum area among all researchers in the sample as the denominator in the standardization procedure.

A researcher's $A_{rel}$ therefore represents her/his citation *mass*, but this value still requires correction for individual opportunity (time since first publication, *t* = current year–$Y_1$) to compare researchers at different stages of their career. This is where career gaps can be taken into account explicitly for any researcher in the sample by subtracting $a_i$ = the total cumulative time absent from research (e.g., maternity or paternity leave, sick leave, secondment, etc.) for individual *i* from *t*, such that an individual's career gap-corrected $t'_i = t_i - a_i$. We therefore constructed another linear model of the form $A_{rel} \sim \gamma + \theta \log_e t$ across all researchers in the sample, and took the residual ($\varepsilon$) of an individual researcher's $A_{rel}$ from the predicted relationship as a metric of citation performance relative to the rest of the researchers in that sample (S2 Fig). This residual $\varepsilon$ allows us to rank all individuals in the sample from highest (highest citation performance relative to opportunity and the entire sample) to lowest (lowest citation performance relative to opportunity and the entire sample). Any researcher in the sample with a positive $\varepsilon$ is considered to be performing above expectation (relative to the group and the time since first publication), and those with a negative $\varepsilon$ fall below expectation. This approach also has the advantage of fitting different linear models to subcategories within a sample to rank researchers within their respective groupings (e.g., such as by gender; S3 Fig). An R code function to produce the index and its variants using a sample dataset is available from github.com/cjabradshaw/EpsilonIndex.

## Discipline standardization

Each sampled discipline has its own citation characteristics and trends [17], so we expect that the distribution of residuals ($\varepsilon$) within each discipline to be meaningful only for that discipline's sample. We therefore endeavoured to scale ('normalize') the results such that researchers in different disciplines could be compared objectively and more fairly.

We first scaled the $A_{\text{rel}}$ within each discipline by dividing each $i$ researcher's $A_{\text{rel}}$ by the sample's root mean square:

$$A'_{\text{rel}_i} = \frac{A_{\text{rel}_i}}{\sqrt{\dfrac{\sum_{i=1}^{n} A_{\text{rel}_i}}{n-1}}}$$

where $n$ = the total number of researchers in the sample ($n$ = 60). We then regressed these discipline-scaled $A'_{\text{rel}}$ against the $\log_e$ number of years since first publication pooling all sampled disciplines together, and then ranked these scaled residuals ($\varepsilon'$) as described above. Comparison between disciplines is only meaningful when a sufficient sample of researchers from within specific disciplines first have their $\varepsilon$ calculated (i.e., discipline-specific $\varepsilon$), and then each discipline sample undergoes the standardization to create $\varepsilon'$. Then, any sample of researchers from any discipline can be compared directly.

## Results

Despite the considerable variation in citation metrics among researchers and disciplines, there was broad consistency in the strength of the relationships between citation mass ($A_{\text{rel}}$) and $\log_e$ years publishing ($t$) across disciplines (Fig 1), although the geology (GEO) sample had the poorest fit ($^{\text{ALL}}R^2$ = 0.43; Fig 1). The distribution of residuals $\varepsilon$ for each discipline revealed substantial difference in general form and central tendency (Fig 2), but after scaling, the distributions of $\varepsilon'$ became aligned among disciplines and were approximately Gaussian (Shapiro-Wilk normality tests; see Fig 2 for test values).

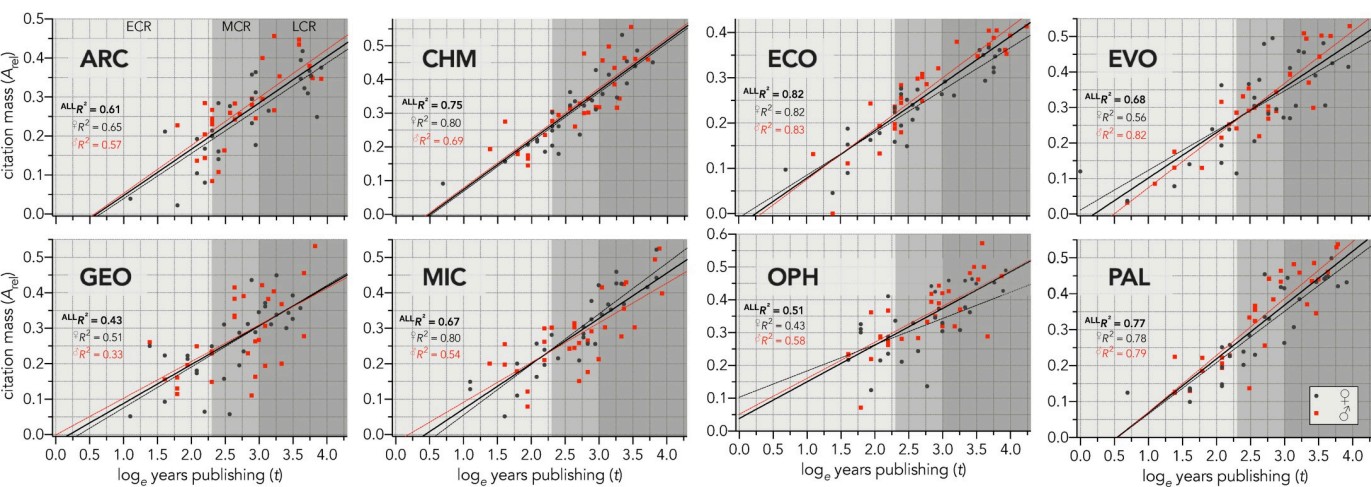

**Fig 1. Citation mass relative to years since first publication.** Relationship between a researcher's citation mass ($A_{\text{rel}}$; area under the citation frequency–value curve—see S2 Fig) and $\log_e$ years ($t$) since first peer-reviewed publication ($Y_1$) for eight disciplines (ARC = archaeology, CHM = chemistry, ECO = ecology, EVO = evolution and development, GEO = geology, MIC = microbiology, OPH = ophthalmology, PAL = palaeontology) comprising 60 researchers each (30 ♀, 30 ♂) in three different career stages: Early career researcher (ECR), mid-career researcher (MCR), and late career researcher (LCR). The fitted lines correspond to the entire sample (solid black), women only (dashed black), and men only (dashed red). Information-theoretic evidence ratios for all relationships > 180; adjusted $R^2$ for each relationship shown in each panel.

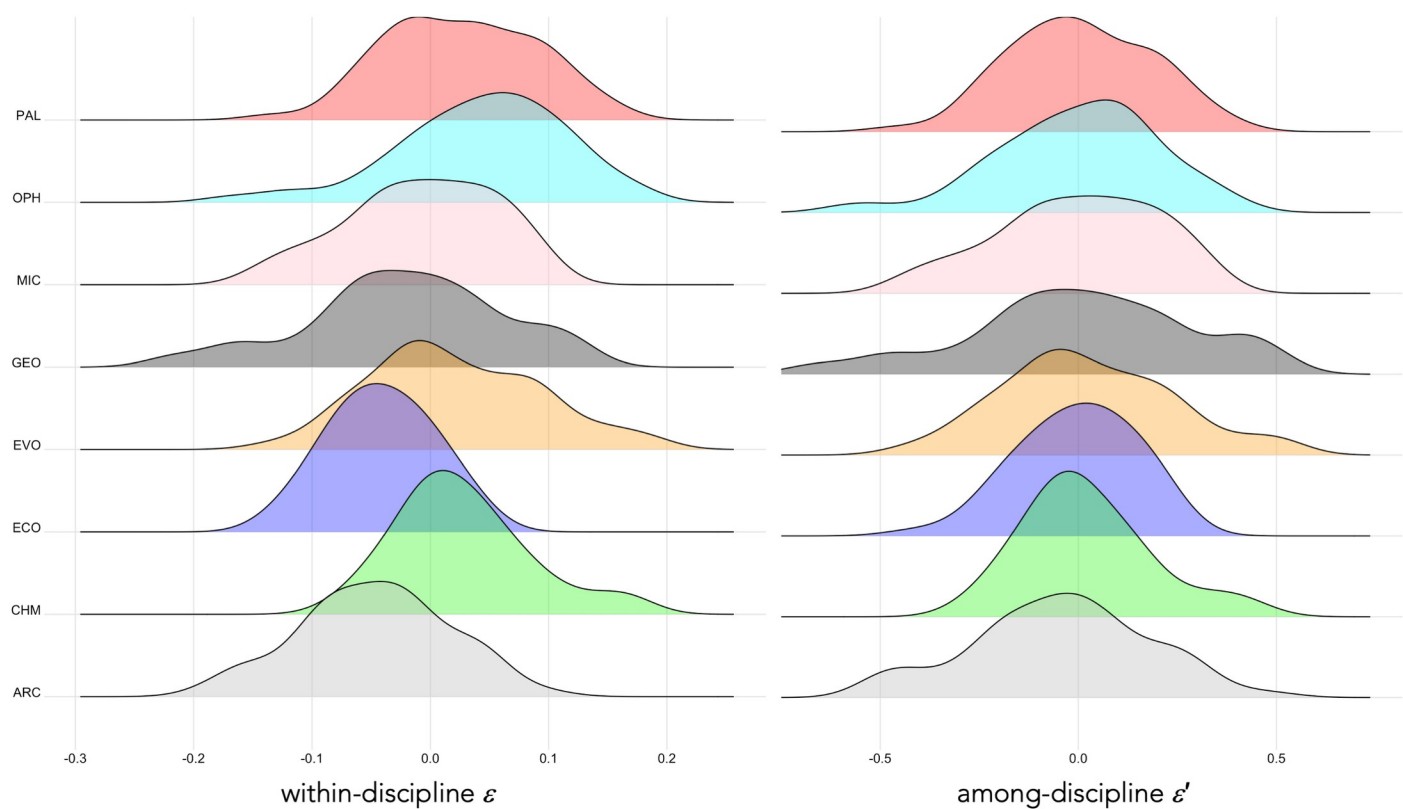

**Fig 2. Within-discipline residuals from the relationship between citation mass and years since first publication.** Left panel: Distribution of within-discipline residuals ($\varepsilon$) of the relationship between $A_{rel}$ and $\log_e$ years publishing ($t$) by discipline (ARC = archaeology, CHM = chemistry, ECO = ecology, EVO = evolution and development, GEO = geology, MIC = microbiology, OPH = ophthalmology, PAL = palaeontology), each comprising 60 researchers (30 ♀, 30 ♂). Right panel: Distribution of among-discipline residuals ($\varepsilon'$) of the relationship between $A'_{rel}$ (scaled) and $t$ by discipline. All $A'_{rel}$ distributions are approximately Gaussian according to Shapiro-Wilk normality tests (ARC: $W = 0.985$, $p = 0.684$; CHM: $W = 0.961$, $p = 0.051$; ECO: $W = 0.980$, $p = 0.409$; EVO: $W = 0.984$, $p = 0.630$; GEO: $W = 0.929$, $p = 0.398$; MIC: $W = 0.971$, $p = 0.170$; OPH: $W = 0.980$, $p = 0.416$; PAL: $W = 0.986$, $p = 0.720$).

After scaling (Fig 3A), the relationship between $\varepsilon'$ and the $m$-quotient is non-linear and highly variable (Fig 3B), meaning that $m$-quotients often poorly reflect actual relative performance (and despite the $m$-quotient already being 'corrected' for $t$, it still increases with $t$; S4 Fig). For example, there are many researchers whose $m$-quotient $< 1$, but who perform above expectation ($\varepsilon' > 0$). Alternatively, there are many researchers with an $m$-quotient of up to 2 or even 3 who perform below expectation ($\varepsilon' < 0$). Once the $m$-quotient $> 3$, $\varepsilon'$ reflects above-expectation performance for all researchers in the example sample (Fig 3B). The corresponding $\varepsilon'$ indicate a more uniform spread by gender and career stage (Fig 3C) than do $m$-quotients (Fig 3D). Further, the relationship between $h$-index and t (from which the $m$-quotient is derived is neither homoscedastic nor Normal (S5–S12 Figs). Another advantage of $\varepsilon'$ *versus* the $m$-quotient is that the former has a threshold ($\varepsilon' = 0$) above which researchers perform above expectation and below which they perform below expectation, whereas the $m$-quotient has no equivalent threshold. Further, the $m$-quotient tends to increase through one's career, whereas $\varepsilon'$ is more stable. There is still an increase in $\varepsilon'$ during late career relative to mid-career, but this is less pronounced that that observed for the $m$-quotient (Fig 4).

Examining the ranks derived from $\varepsilon'$ across disciplines, genders and career stage (Fig 5), bootstrapped median ranks overlap for all sampled disciplines (Fig 5A), but there are some notable divergences between the genders across career stage (Fig 5B). In general, women

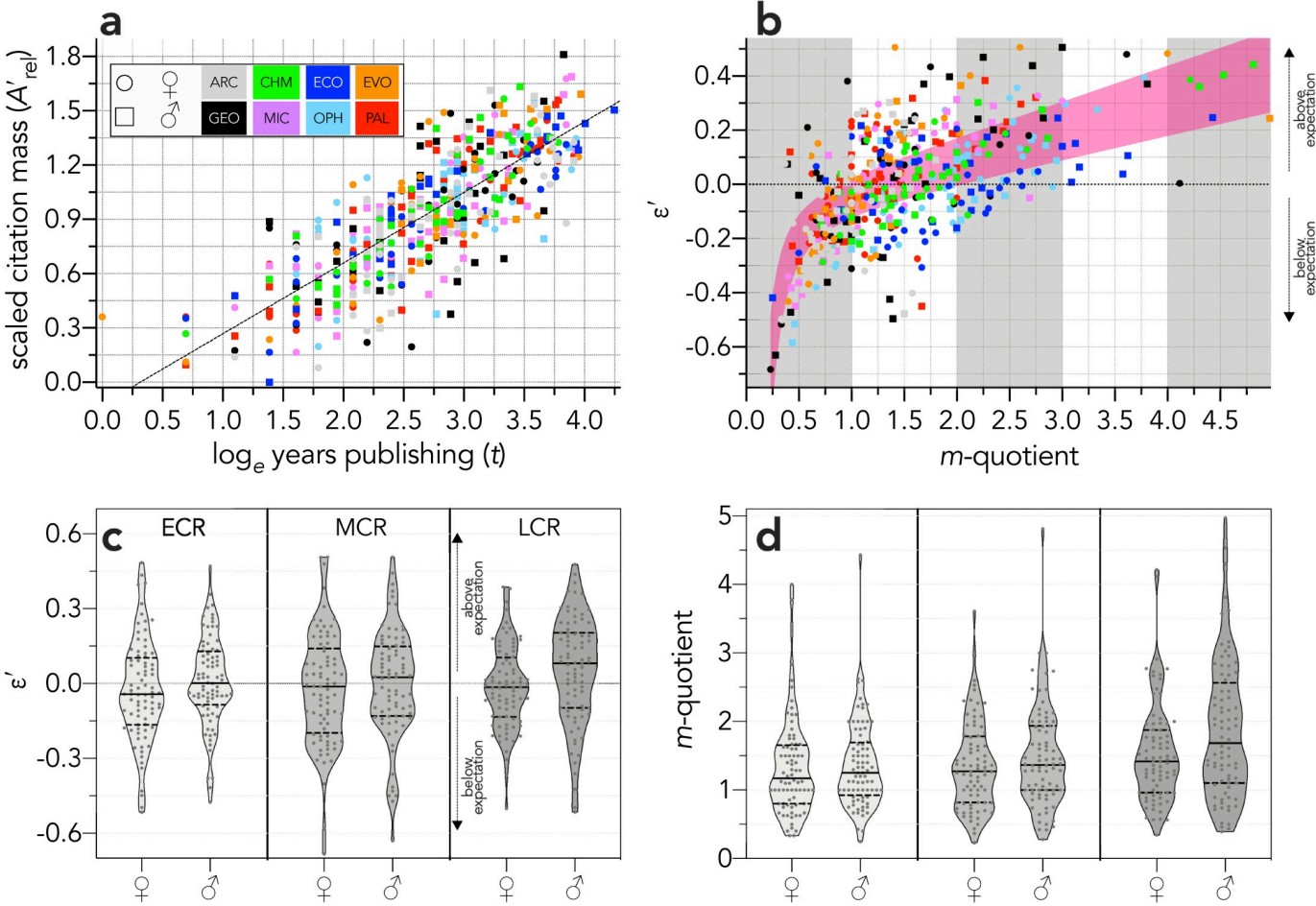

**Fig 3. ε-index versus m-quotient.** (**a**) Relationship between scaled citation mass ($A'_{rel}$) and $\log_e$ years publishing ($t$) for 480 researchers in eight different disciplines (ARC = archaeology, CHM = chemistry, ECO = ecology, EVO = evolution and development, GEO = geology, MIC = microbiology, OPH = ophthalmology, PAL = palaeontology) comprising 60 researchers each (30 ♀, 30 ♂). (**b**) Relationship between the residual of $A'_{rel} \sim \log_e t$ ($\varepsilon'$) and the $m$-quotient for the same researchers (pink shaded area is the 95% confidence envelope of a heat-capacity relationship of the form: $y = a + bx + c/x^2$, where $a$ = -0.17104 –-0.0875; $b$ = 0.0880–0.1318, and $c$ = -0.0423 –-0.0226). (**c**) Truncated violin plots of $\varepsilon'$ by gender and career stage (ECR = early career researcher, MCR = mid-career researcher, LCR = late-career researcher). When $\varepsilon' < 0$, the researcher's citation rank is below expectation relative to her/his peers in the sample; when $\varepsilon' > 0$, the citation rank is greater than expected relative to her/his peers in the sample (dashed lines = quartiles; solid lines = medians). (**d**) Truncated violin plot of the $m$-quotient by gender and career stage.

ranked slightly below men in all career stages, although the bootstrapped median ranks overlap among early and mid-career researchers. However, the median ranks for late-career women and men do not overlap (Fig 5B), which possibly reflects the observation that senior academic positions in many disciplines are dominated by men [26–28], and that women tend to receive fewer citations than men at least in some disciplines, which often tends to compound over time [29–32]. The ranking based on the $m$-quotient demonstrates the disparity among disciplines (Fig 5C), but it is perhaps somewhat more equal between the genders (Fig 5D) compared to the $\varepsilon'$ rank (Fig 5B), despite the higher variability of the $m$-quotient bootstrapped median rank.

However, calculating the scaled residuals across all sampled disciplines for each gender separately, and then combining the two datasets and recalculating the rank (producing a gender-'debiased' rank) effectively removed the gender differences (Fig 6).

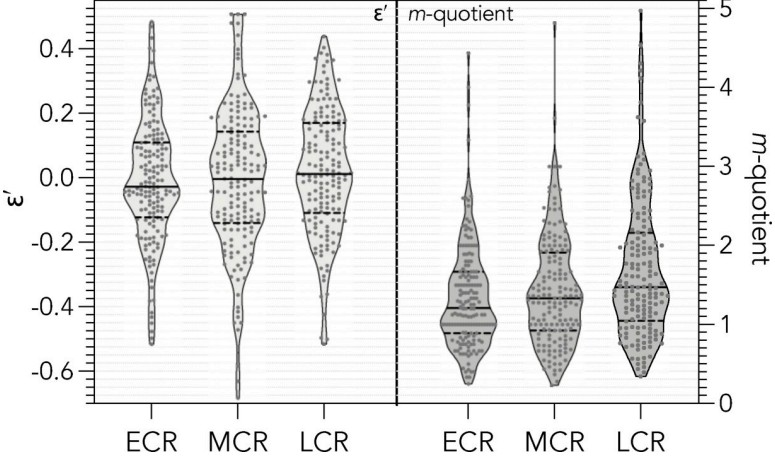

**Fig 4. Career-stage differences in the $\varepsilon'$-index and $m$-quotient.** Violin plots of scaled residuals ($\varepsilon'$) and $m$-quotient across all eight disciplines relative to career stage (ECR = early career; MCR = mid-career; LCR = late career). Treating career stage as an integer in a linear model shows no difference among stages for $\varepsilon'$ ($p = 0.205$), but there is evidence for a career stage effect for the $m$-quotient ($p = 0.000073$). Likewise, treating career stage as an ordinal factor (ECR < MCR < LCR) in a linear model shows no difference among stages for $\varepsilon'$ (MCR: $p = 0.975$; LCR: $p = 0.205$), but there is evidence for a divergence of LCR for the $m$-quotient (MCR: $p = 0.388$; LCR: $p = 0.000072$).

## Discussion

Todeschini and Baccini [33] recommended that the ideal author-level indicator of citation performance should (*i*) have an unequivocal mathematical definition, (*ii*) be easily computed from available data (for a detailed breakdown of implementation steps and the R code function, see github.com/cjabradshaw/EpsilonIndex; we have also provided a user-friendly app available at cjabradshaw.shinyapps.io/epsilonIndex that implements the code and calculates the index with user-provided citation data), (*iii*) balance rankings between more experienced and novice researchers (*iv*) while preserving sensitivity to the performance of top researchers, and (*iv*) be sensitive to the number and distribution of citations and articles. Our new $\varepsilon$-index not only meets these criteria, it also adds the ability to compare across disciplines by using a simple scaling approach, and can easily be adjusted for career gaps by subtracting research-inactive periods from the total number of years publishing (*t*). In this way, the $\varepsilon$-index could prove invaluable as we move toward greater interdisciplinarity, where tenure committees have had difficulty assessing the performance of candidates straddling disciplines [34, 35]. The $\varepsilon$-index does not ignore high-citation papers, but neither does it overemphasize them, and it includes an element of publication frequency ($i_{10}$) while simultaneously incorporating an element of 'quality' by including the $h$-index.

Like all other existing metrics, the $\varepsilon$-index does have some disadvantages in terms of not correcting for author contribution—such as the $h_m$-index [10] or $g_m$-index [11]—even though these types of metrics can be cumbersome to calculate. Early career researchers who have published but have yet to be cited will not yet be able to calculate their $\varepsilon$-index, as they will not have an $h$-index score, so would require different types of assessment. Another potential limitation is that the $\varepsilon$-index alone does not correct for any systemic gender biases associated with the many reasons why women tend to be cited less than men [26–32, 36], but it does easily allow an assessor to benchmark any subset of researchers (e.g., women-only or men-only) to adjust the threshold accordingly. Thus, women can be compared to other women and ranked accordingly such that the ranks are more comparable between these two genders. Alternatively, dividing the genders and benchmarking them separately followed by a combined re-ranking (Fig 6) effectively removes the gender bias in the $\varepsilon$-index, which is difficult or

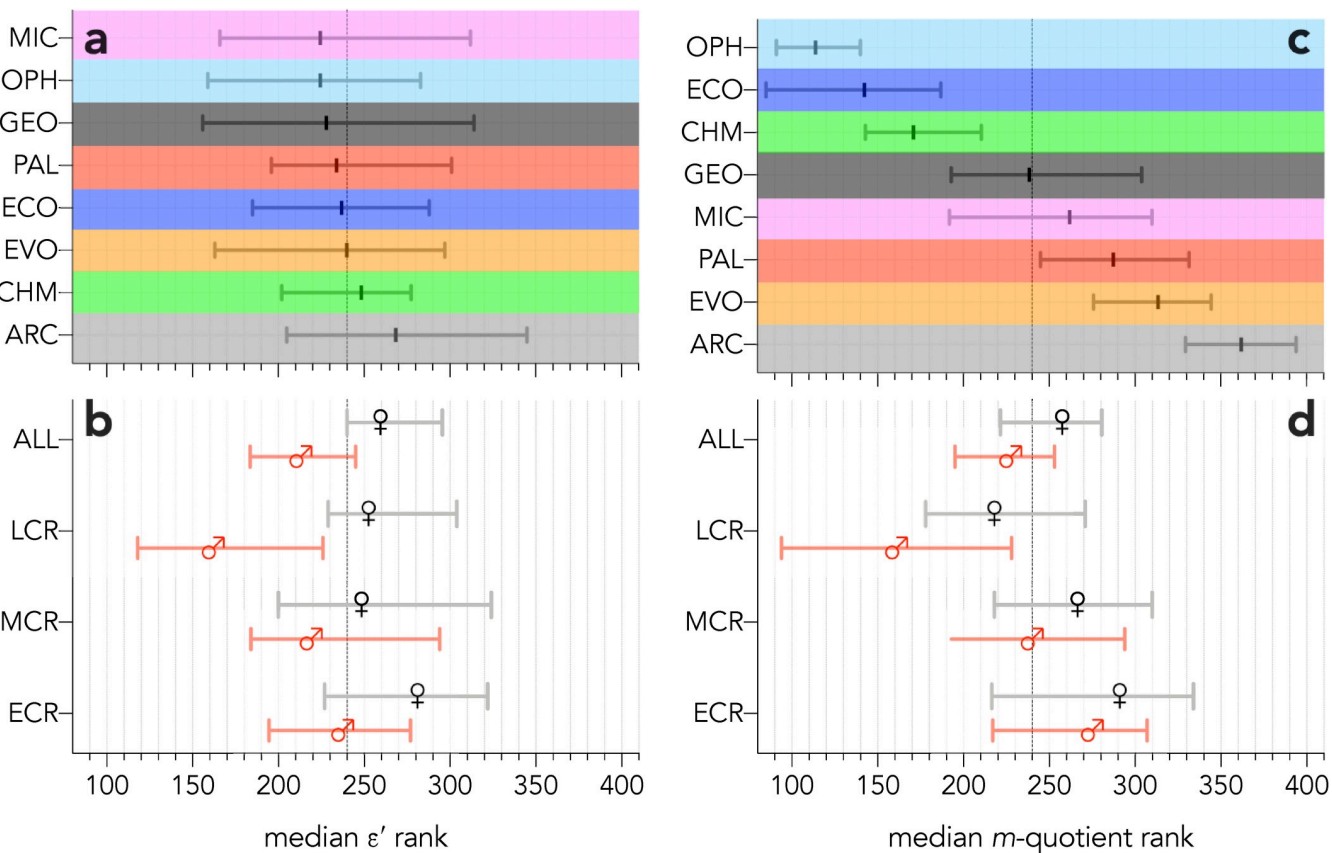

**Fig 5. Gender differences in the $\varepsilon'$-index and $m$-quotient.** (a) Bootstrapped (10,000 iterations) median ranks among the eight disciplines examined (ARC = archaeology, CHM = chemistry, ECO = ecology, EVO = evolution and development, GEO = geology, MIC = microbiology, OPH = ophthalmology, PAL = palaeontology) based on the scaled residuals ($\varepsilon'$). (b) Bootstrapped $\varepsilon'$ ranks by gender and career stage (ECR = early career researcher, MCR = mid-career researcher, LCR = late-career researcher). (c) Bootstrapped (10,000 iterations) median ranks among the eight disciplines based on the $m$-quotient. (d) Bootstrapped $m$-quotient ranks by gender and career stage. The vertical dashed line in all panels indicates the mid-way point across the entire sample (480 ÷ 2 = 240).

impossible to do with other ranking metrics. We certainly advocate this approach when assessing mixed-gender samples (the same approach could be applied to other subsets of researchers deemed *a priori* to be at a disadvantage).

The $\varepsilon$-index also potentially suffers from the requirement of the constituent citation data upon which it is based being accurate and up-to-date [37, 38]. It is therefore important that users correct for obvious errors when compiling the four required data to calculate the $\varepsilon$-index ($i_{10}$, $h$-index, $c_m$, $t$). This could include corrections for misattributed articles, start year, or even $i_{10}$. In some cases, poorly maintained Google Scholar profiles might exclude certain researchers from comparative samples. Regardless, should an assessor have access to potentially more rigorous citation databases (e.g., Scopus), the $\varepsilon$-index can still be readily calculated, although within-sample consistency must be maintained for the ranks to be meaningful. Nonetheless, because the index is relative and scaled, the relative rankings of researchers should be maintained irrespective of the underlying database consulted to derive the input data. We also show that the distribution of the $\varepsilon$-index is relatively more Gaussian and homoscedastic than the time-corrected $m$-quotient, with the added advantage of identifying a threshold above and below which individuals are deemed to be performing better or worse than expected relative to their sample peers. While there are potentially subjective rules of thumb for thresholds to be

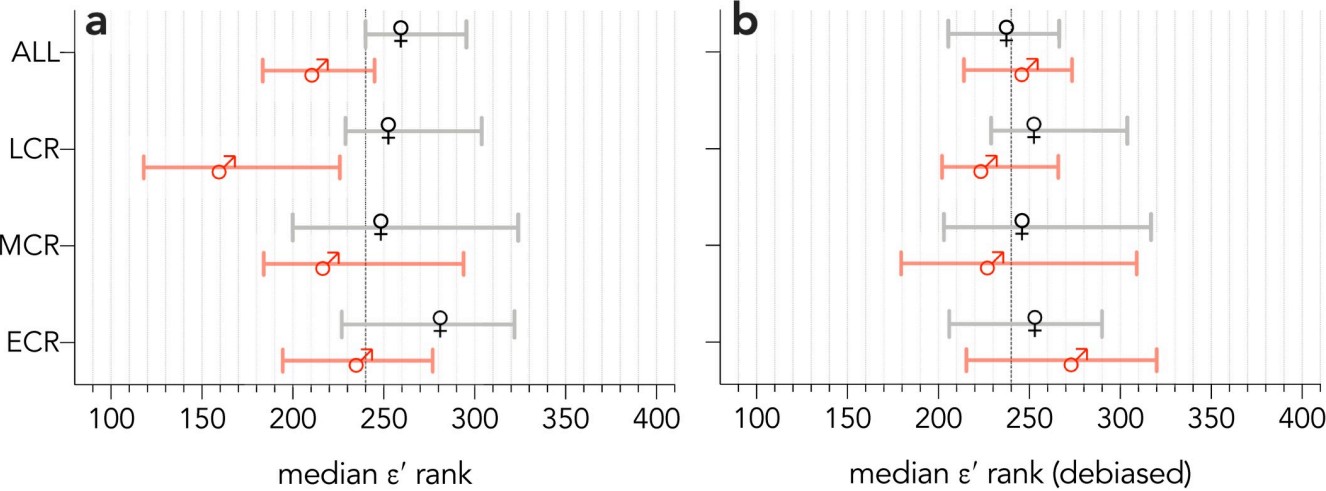

**Fig 6. Gender differences in the $\varepsilon'$-index and gender-debiased $\varepsilon'$-index.** (**a**) Bootstrapped (10,000 iterations) $\varepsilon'$ ranks by gender and career stage (ECR = early career researcher, MCR = mid-career researcher, LCR = late-career researcher); (**b**) bootstrapped debiased (i.e., calculating the scaled residuals for each gender separately, and then ranking the combined dataset) $\varepsilon'$ ranks by gender and career stage.

applied to the *m*-quotient, the residual nature of the $\varepsilon$-index makes it a more objective metric for assessing relative rank, and the $\varepsilon$-index is less-sensitive than the *m*-quotient regarding the innate rise of ranking as a researcher progresses through her/his career (Fig 4).

We reiterate that while the $\varepsilon$-index is an advance on existing approaches to rank researchers according to their citation history, a single metric should never be the sole measure of a researcher's productivity or potential [39]. Nonetheless, the objectivity, ease of calculation, and flexibility of its application argue that the $\varepsilon$-index is a needed tool in the quest to provide fairer and more responsible [39, 40] initial appraisals of a researcher's publication performance.

## Supporting information

**S1 Fig. Citation frequency *versus* citation value.** Relationship between $\log_e$ citation frequency (*y*) and $\log_e$ citation value (*x*) for 60 researchers within the discipline of ophthalmology. Each light grey, dashed line is the linear (on the $\log_e$-$\log_e$ scale) fit for each individual researcher. The area under the fitted line ($A_{rel}$) is shown for individual 32 (ID32; red horizontal hatch) and individual 27 (orange vertical hatch).
(TIF)

**S2 Fig. Example citation mass relative to years since first publication.** Relationship between a researcher's citation mass ($A_{rel}$; area under the citation frequency–value curve—see S1 Fig) and $\log_e$ years since first peer-reviewed publication ($Y_1$) for an example sample of 60 microbiology researchers in three different career stages: early career researcher (ECR), mid-career researcher (MCR), and late-career researcher (LCR). The residuals ($\varepsilon$) for each researcher relative to the line of best fit (solid black line) indicate relative citation rank—researchers below this line perform below expectation (relative to the sample), those above, above expectation. Also shown are the lines of best fit for women (black dashed line) and men (red dashed line— see also S3 Fig). Here we have also selected two researchers at random (1 female, 1 male) from each career stage and shown their results in the inset table. The residuals ($\varepsilon$) provide a relative rank from most positive to most negative. Also shown is each of these six researchers' *m*-quotient (*h*-index ÷ number of years publishing).
(TIF)

**S3 Fig. Gender-specific rankings.** Gender-specific researcher ranks *versus* ranks derived from the entire sample (in this case, the microbiology sample shown in S2 Fig). For women who increased ranks when only compared to other women (negative residuals; top panel), the average increase was 1.50 places higher. For women with reduced ranks (positive residuals; top panel), the average was 1.88 places lower. For men who increased ranks when only compared to other men (negative residuals; bottom panel), or who declined in rank (positive residuals; bottom panel), the average number of places moved were both 1.75 for both.
(TIF)

**S4 Fig. *m*-quotient relative to years since first publication.** Relationship between the $m$-quotient and $\log_e$ years publishing ($t$) for 480 researchers in eight different disciplines. There is a weak, but statistically supported positive relationship (information-theoretic evidence ratio = 68.7).
(TIF)

**S5 Fig. Normality and homoscedasticity diagnostics for the archaeology sample.** Residual vs. fitted (**a** & **b**), scale-location (**c** & **d**), and normal quantile-quantile (**e** & **f**) plots for the relationship between $\log_e A_{rel}$ (area under the power-law relationship) and $\log_e t$ (years publishing) used to derive the $\varepsilon$-index (top row), and for the relationship between the h-index and $t$ used to derive the m-quotient (bottom row) for 60 researchers in the discipline of archaeology (ARC). The $A_{rel} \sim \log_e(t)$ relationships show homoscedasticity (i.e., a random pattern in the residual vs. fitted plots, and no trend in the scale-location plots) and a near-Normal distribution (points fall on the expected quantile-quantile line). In contrast, the $h$-index $\sim t$ relationships all show heteroscedasticity (i.e., a 'fan' pattern in the residual vs. fitted plots, and a positive trend in the scale-location plots) and a non-Normal distribution (points diverge considerably more from the expected quantile-quantile line).
(TIF)

**S6 Fig. Normality and homoscedasticity diagnostics for the chemistry sample.** Residual vs. fitted (**a** & **b**), scale-location (**c** & **d**), and normal quantile-quantile (**e** & **f**) plots for the relationship between $\log_e A_{rel}$ (area under the power-law relationship) and $\log_e t$ (years publishing) used to derive the $\varepsilon$-index (top row), and for the relationship between the h-index and $t$ used to derive the m-quotient (bottom row) for 60 researchers in the discipline of chemistry (CHM). The $A_{rel} \sim \log_e(t)$ relationships show homoscedasticity (i.e., a random pattern in the residual vs. fitted plots, and no trend in the scale-location plots) and a near-Normal distribution (points fall on the expected quantile-quantile line). In contrast, the $h$-index $\sim t$ relationships all show heteroscedasticity (i.e., a 'fan' pattern in the residual vs. fitted plots, and a positive trend in the scale-location plots) and a non-Normal distribution (points diverge considerably more from the expected quantile-quantile line).
(TIF)

**S7 Fig. Normality and homoscedasticity diagnostics for the ecology sample.** Residual vs. fitted (**a** & **b**), scale-location (**c** & **d**), and normal quantile-quantile (**e** & **f**) plots for the relationship between $\log_e A_{rel}$ (area under the power-law relationship) and $\log_e t$ (years publishing) used to derive the $\varepsilon$-index (top row), and for the relationship between the h-index and $t$ used to derive the m-quotient (bottom row) for 60 researchers in the discipline of ecology (ECO). The $A_{rel} \sim \log_e(t)$ relationships show homoscedasticity (i.e., a random pattern in the residual vs. fitted plots, and no trend in the scale-location plots) and a near-Normal distribution (points fall on the expected quantile-quantile line). In contrast, the $h$-index $\sim t$ relationships all show heteroscedasticity (i.e., a 'fan' pattern in the residual vs. fitted plots, and a positive trend in the

scale-location plots) and a non-Normal distribution (points diverge considerably more from the expected quantile-quantile line).
(TIF)

**S8 Fig. Normality and homoscedasticity diagnostics for the evolution/development sample.** Residual vs. fitted (**a** & **b**), scale-location (**c** & **d**), and normal quantile-quantile (**e** & **f**) plots for the relationship between $\log_e A_{\text{rel}}$ (area under the power-law relationship) and $\log_e t$ (years publishing) used to derive the $\varepsilon$-index (top row), and for the relationship between the h-index and $t$ used to derive the m-quotient (bottom row) for 60 researchers in the discipline of evolution and development (EVO). The $A_{\text{rel}} \sim \log_e(t)$ relationships show homoscedasticity (i.e., a random pattern in the residual vs. fitted plots, and no trend in the scale-location plots) and a near-Normal distribution (points fall on the expected quantile-quantile line). In contrast, the $h$-index $\sim t$ relationships all show heteroscedasticity (i.e., a 'fan' pattern in the residual vs. fitted plots, and a positive trend in the scale-location plots) and a non-Normal distribution (points diverge considerably more from the expected quantile-quantile line).
(TIF)

**S9 Fig. Normality and homoscedasticity diagnostics for the geology sample.** Residual vs. fitted (**a** & **b**), scale-location (**c** & **d**), and normal quantile-quantile (**e** & **f**) plots for the relationship between $\log_e A_{\text{rel}}$ (area under the power-law relationship) and $\log_e t$ (years publishing) used to derive the $\varepsilon$-index (top row), and for the relationship between the h-index and $t$ used to derive the m-quotient (bottom row) for 60 researchers in the discipline of geology (GEO). The $A_{\text{rel}} \sim \log_e(t)$ relationships show homoscedasticity (i.e., a random pattern in the residual vs. fitted plots, and no trend in the scale-location plots) and a near-Normal distribution (points fall on the expected quantile-quantile line). In contrast, the $h$-index $\sim t$ relationships all show heteroscedasticity (i.e., a 'fan' pattern in the residual vs. fitted plots, and a positive trend in the scale-location plots) and a non-Normal distribution (points diverge considerably more from the expected quantile-quantile line).
(TIF)

**S10 Fig. Normality and homoscedasticity diagnostics for the microbiology sample.** Residual vs. fitted (**a** & **b**), scale-location (**c** & **d**), and normal quantile-quantile (**e** & **f**) plots for the relationship between $\log_e A_{\text{rel}}$ (area under the power-law relationship) and $\log_e t$ (years publishing) used to derive the $\varepsilon$-index (top row), and for the relationship between the h-index and $t$ used to derive the m-quotient (bottom row) for 60 researchers in the discipline of microbiology (MIC). The $A_{\text{rel}} \sim \log_e(t)$ relationships show homoscedasticity (i.e., a random pattern in the residual vs. fitted plots, and no trend in the scale-location plots) and a near-Normal distribution (points fall on the expected quantile-quantile line). In contrast, the $h$-index $\sim t$ relationships all show heteroscedasticity (i.e., a 'fan' pattern in the residual vs. fitted plots, and a positive trend in the scale-location plots) and a non-Normal distribution (points diverge considerably more from the expected quantile-quantile line).
(TIF)

**S11 Fig. Normality and homoscedasticity diagnostics for the ophthalmology sample.** Residual vs. fitted (**a** & **b**), scale-location (**c** & **d**), and normal quantile-quantile (**e** & **f**) plots for the relationship between $\log_e A_{\text{rel}}$ (area under the power-law relationship) and $\log_e t$ (years publishing) used to derive the $\varepsilon$-index (top row), and for the relationship between the h-index and $t$ used to derive the m-quotient (bottom row) for 60 researchers in the discipline of ophthalmology (OPH). The $A_{\text{rel}} \sim \log_e(t)$ relationships show homoscedasticity (i.e., a random pattern in the residual vs. fitted plots, and no trend in the scale-location plots) and a near-Normal distribution (points fall on the expected quantile-quantile line). In contrast, the $h$-index $\sim t$

relationships all show heteroscedasticity (i.e., a 'fan' pattern in the residual vs. fitted plots, and a positive trend in the scale-location plots) and a non-Normal distribution (points diverge considerably more from the expected quantile-quantile line).
(TIF)

**S12 Fig. Normality and homoscedasticity diagnostics for the palaeontology sample.** Residual vs. fitted (**a** & **b**), scale-location (**c** & **d**), and normal quantile-quantile (**e** & **f**) plots for the relationship between $\log_e A_{rel}$ (area under the power-law relationship) and $\log_e t$ (years publishing) used to derive the $\varepsilon$-index (top row), and for the relationship between the h-index and $t$ used to derive the m-quotient (bottom row) for 60 researchers in the discipline of palaeontology (PAL). The $A_{rel} \sim \log_e(t)$ relationships show homoscedasticity (i.e., a random pattern in the residual vs. fitted plots, and no trend in the scale-location plots) and a near-Normal distribution (points fall on the expected quantile-quantile line). In contrast, the $h$-index $\sim t$ relationships all show heteroscedasticity (i.e., a 'fan' pattern in the residual vs. fitted plots, and a positive trend in the scale-location plots) and a non-Normal distribution (points diverge considerably more from the expected quantile-quantile line).
(TIF)

## Acknowledgments

We acknowledge our many peers for their stewardship of their online citation records. We acknowledge the Indigenous Traditional Owners of the land on which Flinders University is built—the Kaurna people of the Adelaide Plains.

## Author Contributions

**Conceptualization:** Corey J. A. Bradshaw, Justin M. Chalker, John A. Long, Justine R. Smith.

**Data curation:** Corey J. A. Bradshaw, Justin M. Chalker, Stefani A. Crabtree, Bart A. Eijkelkamp, John A. Long, Justine R. Smith, Kate Trinajstic, Vera Weisbecker.

**Formal analysis:** Corey J. A. Bradshaw.

**Investigation:** Corey J. A. Bradshaw.

**Methodology:** Corey J. A. Bradshaw.

**Project administration:** Corey J. A. Bradshaw.

**Resources:** Corey J. A. Bradshaw.

**Software:** Corey J. A. Bradshaw.

**Visualization:** Corey J. A. Bradshaw.

**Writing – original draft:** Corey J. A. Bradshaw.

**Writing – review & editing:** Corey J. A. Bradshaw, Justin M. Chalker, Stefani A. Crabtree, Bart A. Eijkelkamp, John A. Long, Justine R. Smith, Kate Trinajstic, Vera Weisbecker.

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
