## [Decision Letter · Decision Letter 0]

10 May 2021

PONE-D-21-03358

A fairer way to compare researchers at any career stage and in any discipline using open-access citation data

PLOS ONE

Dear Dr. Bradshaw,

Thank you for submitting your manuscript to PLOS ONE. After careful consideration, we feel that it has merit but does not fully meet PLOS ONE’s publication criteria as it currently stands. Therefore, we invite you to submit a revised version of the manuscript that addresses the points raised during the review process.

Despite considering the paper interesting, both reviewers have raised a number of methodological concerns (please, note PLOS ONE's publication criterion #3, https://journals.plos.org/plosone/s/criteria-for-publication#loc-3). Some of such concerns might have a deep impact in the presented results (i.e. data source or discipline selection) and, therefore, should be paid special attention in your revision of the manuscript. In addition, Reviewer 2 initial comments might help you to better embed your work in the already huge literature of scientific performance indicators.

We look forward to receiving your revised manuscript.

Kind regards,

Sergi Lozano

Academic Editor

PLOS ONE

Journal Requirements:

2. Please note that PLOS ONE has specific guidelines on software sharing (http://journals.plos.org/plosone/s/materials-and-software-sharing#loc-sharing-software) for manuscripts whose main purpose is the description of a new software or software package. In this case, new software must conform to the Open Source Definition (https://opensource.org/docs/osd) and be deposited in an open software archive. Please see http://journals.plos.org/plosone/s/materials-and-software-sharing#loc-depositing-software for more information on depositing your software.

Reviewers' comments:

Reviewer's Responses to Questions

**Comments to the Author**

1. Is the manuscript technically sound, and do the data support the conclusions?

Reviewer #1: Partly

Reviewer #2: Yes

2. Has the statistical analysis been performed appropriately and rigorously? 

Reviewer #1: Yes

Reviewer #2: Yes

3. Have the authors made all data underlying the findings in their manuscript fully available?

Reviewer #1: Yes

Reviewer #2: Yes

4. Is the manuscript presented in an intelligible fashion and written in standard English?

Reviewer #1: Yes

Reviewer #2: Yes

5. Review Comments to the Author

Reviewer #1: This is an interesting paper that develops a measure to evaluate scholars' performance across career stages and disciplines. The algorithms and results are easy to understand with cool visualizations. However, I have several concerns about the experiments and evaluations of the proposed measure.

First, the authors focused on eight disciplines when selecting researchers. But the selected disciplines do not seem to cover major disciplines in science. Most of them are subfields in biomedical research. Some important disciplines such as engineering, math and physics, social science, and computer science are missing in the list. Thus the experimental result does not necessarily support the claim that this measure works across all academic disciplines. I would recommend consulting a standard discipline catalog to reduce selection bias, such as the UCSD map of science (defines 13 disciplines): https://journals.plos.org/plosone/article?id=10.1371/journal.pone.0039464. This catalog is also used in a recent paper: https://advances.sciencemag.org/content/7/17/eabb9004.

Second, the algorithm fits a linear line to each researcher based on three data points. This gives the area under the line A_{rel} for each researcher, which is then scaled to the maximum value in the sample (the one with the highest c_m). But why does the researcher with the highest c_m has the highest A_{rel}? Also, with this framework, if I understand the ranking algorithm correctly, there should exist a data point whose A_{rel} equals 1.0 in Fig. S3, but it's missing.

Third, there is no external validation of the measure. The authors did compare the ranking obtained with the proposed measure to that based on the m-quotient (Fig. 3b), but this is not a proper validation. The paper assumes that this measure just works as expected and then is used as a ground truth to evaluate the m-quotients (by stating that "the relationship between ε′ and the m-quotient is non-linear and highly variable, meaning that m-quotients often poorly reflect actual relative performance"). What if it is the other way around --- the m-quotients ranks researchers in a meaningful way, which would then indicate that the proposed measure fits poorly. The paper does show that there is a level of relationships between A_{rel} and log_e(t) across disciplines with reported R^2 at the beginning. But how much R^2 is needed to support a strong correlation and the fitness of the model? I would recommend the authors validate the ranking against external ground truth data, such as the evaluation of researchers from experts via survey.

Fourth, another limitation of this relative measure is that it's sensitive to the samples used in the ranking, especially when comparing researchers across disciplines. Let's imagine a scenario where one needs to compare two scholars (A and B) in two different disciplines. In one case, the peers we choose for A's discipline all perform worse than A; in another case, the peers selected all outperform A. Both conditions have the same samples for B's discipline. However, the ranking between A and B could be very different in the two conditions. Indeed, it is not meaningful and of little practical value to even considering comparing a computer scientist with a biologist in the first place.

I think the paper could be improved based on these suggestions.

Minor issue: Fig. 4 does not prove that "the m-quotient tends to increase through one's career, whereas sigma' is more stable" because the errorbars all seem to overlap with each other.

Reviewer #2: Overall I find very interesting and well-written this article. It is well structured, conceived, and executed.

I must confess that yet another article about h-index variants is not the road that the Bibliometrics community is looking for. A lot of (unused) variants have been published and, at the end of the day, only few of them add something to the discussion. In practical terms (availability of the indicator), only h-index is really used (g-index is rarely used in research evaluation in most countries).

We need to separate the advancements of Bibliometrics, on the one hand, and the use of indicators for research evaluation, on the other. Despite the clear intersections among these fields, we find significant differences in their approaches and interests.

Therefore, I would recommend authors to emphasize the limitations of previous indicators (and such uncovered things that research evaluation tasks still need) in greater detail, in order to justify properly the new proposal. Without a clear description of literature (and professional use of these indicators) gaps, new proposals feel incomplete.

I find excessive the use of the term “fair”, not only in the title but also throughout the text. It is somewhat subjective, and no fair indicator exists. Moreover, I recommend linking strongly the proposal of new indicator with responsible indicators and responsible research movements.

Please find below some minor comments, suggestions and recommendations oriented to make stronger the proposal.

Among the many disadvantages of the h-index (I fully agree with most of them), I do not find its accumulative nature as a limitation, as long as evaluators use it as wisely as possible. The problem lies with the poor use of the indicator. Obviously a person with 50 years may have more experience and years worked than a 25 years old person. That not makes 'number of years working' a bad indicator itself, it is just incomplete if we want to measure applicant skills.

Despite Google Scholar is free to access, data cannot be massively exported. No API exists, and this database shows some limitations (information noise, duplicates, errors, etc.). Google Scholar Profiles is a filter of Google Scholar, which depends on the author to create the profile accurately. These points should be discussed as it is the database used as a test-bed. Authors include some comments about it, but I believe they need to make stronger the reason to use this database and, later, how we can move to other databases in order to extrapolate the indicator to other controlled environments. If the indicator can only be operated with Google Scholar, it is a limitation.

“The entire approach we present here assumes that each researcher’s Google Scholar profile is accurate, up-to-date, and complete.”

This is a dangerous approach. Real life shows us that profiles are noisy, with errors (some of them on purpose). It is clear that here the important thing is to test the statistical nature of the indicator. However, the sensitive of the indicator to the nature of the database in real conditions can add robustness to this proposal.

Why authors selected these specific disciplines? Why the number of researchers is equal? The demography of these disciplines is not equal.

I understand the research design and the underlying reasons. However, again, the real conditions of the database should be acknowledged, and decisions should be strongly justified. With the results obtained I cannot be sure if the indicator would be useful for other disciplines and, then, generalizing the strengths of the indicator.

“we did not intend for sampling to be a definitive comment about the performance of particular researchers, nor did we mean for each sample to represent an entire discipline”

While I understand this point, this is important, as authors are trying to operate an indicator with a particular dataset. Biases of this dataset can be inherited in the conclusions achieved. If the sample does not represent the entire discipline, then how I can infer its usefulness to this discipline?

“peer-reviewed article published in a recognized scientific journal”

What a recognized scientific journal is for authors in the context of google scholar profiles?

“For the designation of Y1, we excluded any reports, chapters, books, theses or other forms of publication that preceded the year of the first peer-reviewed article; however, we included citations from the former sources in the researcher’s i10, h, and cm.”

I disagree with this procedure. I do not see justifiable to exclude book chapters as document, and later include their citations, it can introduce citation biases. Please justify this decision.

All figures performed are of excellent quality and are very informative. The data segregation according to gender is so interesting and adds new debates and discussions. I congratulate authors for this effort in data visualization.

6. PLOS authors have the option to publish the peer review history of their article (what does this mean?). If published, this will include your full peer review and any attached files.

Reviewer #1: No

Reviewer #2: No

---

## [Author Response · Author response to Decision Letter 0]

18 Jul 2021

Reviewer #1

This is an interesting paper that develops a measure to evaluate scholars' performance across career stages and disciplines. The algorithms and results are easy to understand with cool visualizations. However, I have several concerns about the experiments and evaluations of the proposed measure.

First, the authors focused on eight disciplines when selecting researchers. But the selected disciplines do not seem to cover major disciplines in science. Most of them are subfields in biomedical research. Some important disciplines such as engineering, math and physics, social science, and computer science are missing in the list. Thus the experimental result does not necessarily support the claim that this measure works across all academic disciplines. I would recommend consulting a standard discipline catalog to reduce selection bias, such as the UCSD map of science (defines 13 disciplines):

https://journals.plos.org/plosone/article?id=10.1371/journal.pone.0039464. This catalog is also used in a recent paper: https://advances.sciencemag.org/content/7/17/eabb9004.

RESPONSE #1: Had our goal been to test the universality of our proposed metric across all disciplines, we would have followed a procedure similar to the one the reviewer proposes. But this was not our goal. Rather, we aimed to choose a sample of widely divergent disciplines in terms of citation trends, that had an equal number of men and women in each sample, as well as an equal number of researchers in each of the three career stages we identified. 

To achieve such a gender- and career-stage-balanced sample required knowledge about the discipline and a good deal of experience in ranking researchers therein. That was why each co-author was chosen from a different discipline, and not across the entire array of disciplines. We also aimed to have as much gender and career-level diversity in our authorship team.

We disagree that the sample disciplines were primarily “biomedical”; in fact, only one was strictly in the biomedical and clinical sciences (ophthalmology), three were in biological sciences (ecology, evolution/development, microbiology), two were in earth sciences (geology, palaeontology), one was in chemical sciences (chemistry), and one in history, heritage and archaeology (archaeology). The discipline categories we provide here are the official Field of Research major categories from the Australian Bureau of Statistics (www.abs.gov.au/statistics/classifications/australian-and-new-zealand-standard-research-classification-anzsrc/2020).

Did we cover a sufficient spread of disciplines according to citation trends? Yes. According to InCites Essential Science Indicators™, highly cited papers range from 24.56 to 4.85 cites/paper across 22 disciplines — our sample covered 83.4% of that range: molecular biology/genetics 24.56; chemistry 16.30; environment/ecology 14.64, geosciences 14.04, clinical medicine 13.72, social sciences 8.12.

More importantly, we never claimed that our metric applies universally across “all disciplines”. Throughout the text where we stated “all disciplines”, this was explicitly with references to “all [sampled] disciplines”. To clarify, we have now added the word ‘sampled’ where appropriate.

Second, the algorithm fits a linear line to each researcher based on three data points. This gives the area under the line A_(rel) for each researcher, which is then scaled to the maximum value in the sample (the one with the highest c_m). But why does the researcher with the highest c_m has the highest A_(rel)?

RESPONSE #2: We concede that we had not adequately explained the standardisation procedure. The algorithm predicts a series of slices across the triangle made by the slope and intercept estimated from the fit between the 3 points in the citation frequency by citation value graph (e.g., previously Fig. S2; now Fig. S1). The citation mass — the sum of these slices (i.e., predicted loge citation frequency for incrementing values of loge citation value) — is then standardised by taking the sum and dividing it by the number of slices multiplied by the maximum of loge cm across all researchers. The main reason we did this is explained more in the following response (Response #4), but we have now added some clarifying text in the Methods:

“Here, the sum of the predicted y derived from incrementing values of x (here in units of 0.05) using α ^ and β ^ is divided by the product of cm and the number of incremental x values. This implies all areas were scaled to the maximum in the sample, but avoids the problem of truncating variances near a maximum of 1 had we used the maximum area among all researchers in the sample as the denominator in the standardization procedure.”

Also, with this framework, if I understand the ranking algorithm correctly, there should exist a data point whose A_(rel) equals 1.0 in Fig. S3, but it's missing.

RESPONSE #3: Because we do not standardise each researcher’s triangle area via division by the largest triangle area in the sample, the citation masses do not have a maximum of 1. The principal reason for avoiding this is to prevent truncation of variances closer to the ‘extreme’ values near 1. This could violate the homoscedasticity property of the ε index, which is a problem with the m-quotient (see Response #4) 

Third, there is no external validation of the measure. The authors did compare the ranking obtained with the proposed measure to that based on the m-quotient (Fig. 3b), but this is not a proper validation. The paper assumes that this measure just works as expected and then is used as a ground truth to evaluate the m-quotients (by stating that "the relationship between ε′ and the m-quotient is non-linear and highly variable, meaning that m-quotients often poorly reflect actual relative performance"). What if it is the other way around --- the m-quotients ranks researchers in a meaningful way, which would then indicate that the proposed measure fits poorly. The paper does show that there is a level of relationships between A_(rel) and log_e(t) across disciplines with reported R^2 at the beginning. But how much R^2 is needed to support a strong correlation and the fitness of the model? I would recommend the authors validate the ranking against external ground truth data, such as the evaluation of researchers from experts via survey.

RESPONSE #4: We had originally designed an internal ‘evaluation’ of the researcher lists we compiled for each discipline, where each co-author would subjectively rank the researchers in their respective discipline according to their own qualitative and quantitative criteria. This is yet another reason we kept the sample of disciplines to those in which our eight co-authors were most experienced. 

However, we ultimately abandoned this line of inquiry because there was really no way to guarantee any standardisation or ‘truth’ in the subjective rankings. Whether these subjective rankings were well-correlated or not with ε does not insinuate that ε is any worse or better than existing metrics per se. This is, after all, the entire aim of our study — to provide a better metric than what is currently available. We never insinuated that it reflects absolute reality (whatever that might mean, in the nebulous world of researcher-ranking algorithms). Because everyone uses a different set of criteria to rank researchers, such a “ground truth” ends up being none at all.

From a statistical perspective, there is no threshold beyond which the R2 between citation mass (Arel) and loge years publishing (t) becomes ‘acceptable’ — it is a range (i.e., higher is better, lower is worse). However, we can show that the relationship between the h-index and years publishing (the quotient of which is the m-index) violates several statistical assumptions, whereas ε does not. We did not include these statistical assumption checks in the first submission, but realise now that they are useful for justifying our approach.

The following plots, which we have now added to the Supplementary Material (Fig. S5–S12), show the residual vs. fitted, scale-location, and normal quantile-quantile plots for the relationship between Arel and loge(t) (top rows) and between h-index and t (bottom rows) for all eight disciplines — the latter relationship represents the m-quotient (h-index ÷ t):

In all disciplines, the Arel ~ loge(t) relationships show homoscedasticity (i.e., a random pattern in the residual vs. fitted plots, and no trend in the scale-location plots) and a near-Normal distribution (points fall on the expected quantile-quantile line).

In contrast, the h-index ~ t relationships all show heteroscedasticity (i.e., a ‘fan’ pattern in the residual vs. fitted plots, and a positive trend in the scale-location plots) and a non-Normal distribution (points diverge considerably more from the expected quantile-quantile line).

In other words, our ε index does not violate base statistical assumptions in its derivation like the m-quotient does. This demonstration is, however, intuitive given that one can clearly see how the m-quotient is truncated at lower values and its variance inflates at larger values relative to the non-bounded ε index (Fig. 3b).

Importantly — and this is really an essential characteristic of the ε index relative to the m-quotient — the latter has no intrinsic threshold to which one can compare relative performance. On the contrary, the ε index explicitly defines a mid-point (value = 0) above which researchers are relatively higher performers, and below which they are relatively poorer performers. The m-index does not have this highly useful characteristic.

Fourth, another limitation of this relative measure is that it's sensitive to the samples used in the ranking, especially when comparing researchers across disciplines. Let's imagine a scenario where one needs to compare two scholars (A and B) in two different disciplines. In one case, the peers we choose for A's discipline all perform worse than A; in another case, the peers selected all outperform A. Both conditions have the same samples for B's discipline. However, the ranking between A and B could be very different in the two conditions. Indeed, it is not meaningful and of little practical value to even considering comparing a computer scientist with a biologist in the first place.

RESPONSE #5: We disagree that this is a weakness and strongly argue the opposite — this is a particular strength of the ε index.

First, we disagree that comparing researchers in different disciplines is not meaningful, because people do it all the time. Whether it is for ranking job applicants, or members of a multidisciplinary centre, the utility is without question.

The lead author (CJAB) has done exactly this sort of comparison on many occasions, having interviewed ecologists, mathematicians, physicists, economists, and engineers for the same position (all requiring mathematical skills for an ecological application). He has also been requested to rank the Chief Investigators in Centres of Excellence spanning the sciences and humanities, as well as rank applicants for nationally competitive grants (e.g., the Australian Research Council and the New Zealand Marsden Fund). Indeed — this necessity was one of the underlying rationales for developing the ε index in the first place.

This justification aside, the reviewer’s example is not how we proposed that the interdisciplinary comparisons should be done, and how the discipline-standardised ε′ index is calculated.

Consider the example provided above. If one wishes to compare researcher A to researcher B, the first step is to accumulate the data for a sample of A’s colleagues in A’s same discipline, and then do the same for a sample of B’s colleagues in B’s discipline. Once the ε index is calculated for discipline A and B separately, the standardisation is applied to each discipline in turn. Then, and only then, can one use A’s ε′ and compare it to B’s ε′. In other words, the within-discipline standardisation of ε to ε′ ensures that A’s and B’s indices are on the same scale.

Given this confusion, we have added the following text in the ‘Discipline standardization’ section of the Methods to clarify:

“Comparison between disciplines is only meaningful when a sufficient sample of researchers from within specific disciplines first have their ε calculated (i.e., discipline-specific ε), and then each discipline sample undergoes the standardization to create ε′. Then, any sample of researchers from any discipline can be compared directly.”

Minor issue: Fig. 4 does not prove that "the m-quotient tends to increase through one's career, whereas sigma' is more stable" because the errorbars all seem to overlap with each other.

RESPONSE #6: Generally speaking, quantile derivation of confidence bounds cannot be substituted for actual statistical tests. It was our fault not to have included such tests, which we now supply in the caption of Fig. 4.

Treating career stage either as an integer (early = 1, mid = 2, late = 3), or as an ordinal factor, in a linear model clearly shows that ε′ does not differ among career stages, but that the late-career m-quotient in particular diverges statistically from the earlier stages:

career stage as an integer

ε′ index:

 Estimate SE t value Pr(>|t|)

(Intercept) -0.02891 0.02461 -1.175 0.241

stage 0.01446 0.01139 1.269 0.205

m-quotient:

 Estimate SE t value Pr(>|t|)

(Intercept) 1.14857 0.09261 12.403 < 2e-16 ***

stage 0.17152 0.04287 4.001 7.3e-05 ***

career stage as an ordinal factor

ε′ index:

 Estimate SE t value Pr(>|t|)

(Intercept) -0.00987 0.01612 -0.612 0.541

stage2 0.00070 0.02280 0.031 0.975

stage3 0.02891 0.02280 1.268 0.205

m-quotient:

 Estimate SE t value Pr(>|t|)

(Intercept) 1.35259 0.06058 22.328 < 2e-16 ***

stage2 0.07402 0.08567 0.864 0.388 

stage3 0.34304 0.08567 4.004 7.22e-05 ***

We have elected not to include all this detail in the caption, but have provided the Type I error estimates Pr(>|t|) for both the integer- and ordinal factor-based linear models. The new text in the caption of Fig. 4 is now:

“Treating career stage as an integer in a linear model shows no difference among stages for ε′ (p = 0.205), but there is evidence for a career stage effect for the m-quotient (p = 0.000073). Likewise, treating career stage as an ordinal factor (ECR < MCR < LCR) in a linear model shows no difference among stages for ε′ (MCR: p = 0.975; LCR: p = 0.205), but there is evidence for a divergence of LCR for the m-quotient (MCR: p = 0.388; LCR: p = 0.000072).”

Finally, we have replaced Fig. 4 with a violin plot as a better reflection of the distribution and trends in the underlying data.

Reviewer #2

I must confess that yet another article about h-index variants is not the road that the Bibliometrics community is looking for. A lot of (unused) variants have been published and, at the end of the day, only few of them add something to the discussion. In practical terms (availability of the indicator), only h-index is really used (g-index is rarely used in research evaluation in most countries).

RESPONSE #7: We could not agree more, and is exactly why we derived this new, easily calculated, relative, and less-biased metric. Here, it was a delicate balance between ease of calculation and improvement over existing metrics in terms of career stage and gender biases (prevalent in both the h-index and m-quotient as we show). We contend that the main reason most people default to the h-index is because it is provided by most citation-accumulation engines. With a little additional effort, a few more readily available data points can make a world of difference.

Is our ε-index all-encompassing and free of all biases? Does it account for variable contribution in co-authorship? Does it consider non-citation-based metrics of performance? Of course not, and we did not claim that it does. But it is easy to derive, and vastly improves rankings compared to those based on the h-index/m-quotient.

We need to separate the advancements of Bibliometrics, on the one hand, and the use of indicators for research evaluation, on the other. Despite the clear intersections among these fields, we find significant differences in their approaches and interests.

Therefore, I would recommend authors to emphasize the limitations of previous indicators (and such uncovered things that research evaluation tasks still need) in greater detail, in order to justify properly the new proposal. Without a clear description of literature (and professional use of these indicators) gaps, new proposals feel incomplete.

RESPONSE #8: We disagree. This is not a review of the pros and cons of different metrics. That has been done in gory detail many times before [1-7] (note that we have cited all but two of these reviews in the original submission, and have now added them to the Introduction). Our aim was instead to point the reader to these extensive reviews, highlight the remaining problems, and propose one way to account for issues without having to spend too much effort to derive this new index.

 In fact, one bibliometric paper recently published in PLoS One [8] (which we also cited) followed a similar approach to us in these terms.

I find excessive the use of the term “fair”, not only in the title but also throughout the text. It is somewhat subjective, and no fair indicator exists. Moreover, I recommend linking strongly the proposal of new indicator with responsible indicators and responsible research movements.

RESPONSE #9: We have now modified most occasions of ‘fair’ to ‘fairer’, and ‘fairly’ to ‘more fairly’.

While our index does not really fall under the FAIR principles (Findable, Accessible, Interoperable, Reusable) per se (but the article in PLoS One will), we have added a few new citations along the lines of responsible indicators, and adjusted the final paragraph to:

“We reiterate that while the ε-index is an advance on existing approaches to rank researchers according to their citation history, a single metric should never be the sole measure of a researcher’s productivity or potential [9]. Nonetheless, the objectivity, ease of calculation, and flexibility of its application argue that the ε-index is a needed tool in the quest to provide fairer and more responsible [9, 10] initial appraisals of a researcher’s publication performance.”

Among the many disadvantages of the h-index (I fully agree with most of them), I do not find its accumulative nature as a limitation, as long as evaluators use it as wisely as possible. The problem lies with the poor use of the indicator. Obviously a person with 50 years may have more experience and years worked than a 25 years old person. That not makes 'number of years working' a bad indicator itself, it is just incomplete if we want to measure applicant skills.

RESPONSE #10: Agreed. The problem unfortunately is that many people do not use the h-index wisely. A correction for experience is therefore essential.

Despite Google Scholar is free to access, data cannot be massively exported. No API exists, and this database shows some limitations (information noise, duplicates, errors, etc.). Google Scholar Profiles is a filter of Google Scholar, which depends on the author to create the profile accurately. These points should be discussed as it is the database used as a test-bed. Authors include some comments about it, but I believe they need to make stronger the reason to use this database and, later, how we can move to other databases in order to extrapolate the indicator to other controlled environments. If the indicator can only be operated with Google Scholar, it is a limitation.

RESPONSE #11: The lack of an API provided for Google Scholar is a problem, and one of which the lead author has been acutely aware since he developed the ε-index app (https://cjabradshaw.shinyapps.io/epsilonIndex). It would have been ideal to develop a scraper using an API to make the acquisition of the necessary data (i10, h-index, cm, t) automatic. He has even developed the code to do this should Google Scholar ever supply a public API.

But as we stated in the submitted manuscript: “Regardless, should an assessor have access to potentially more rigorous citation databases (e.g., Scopus), the ε-index can still be readily calculated, although within-sample consistency must be maintained for the ranks to be meaningful.”, other databases can be readily used to provide the necessary data. The only limitation, as we stated, is that the same database must be used for all researchers in the sample.

Databases like Scopus are subscription-only, but most universities appear to have subscriptions. Thus, as long as people desiring to construct the ε-index follow this simple rule of consistency, any database can be potentially used.

We have updated the relevant section along these lines:

“The ε-index also potentially suffers from the requirement of the constituent citation data upon which it is based being accurate and up-to-date [11, 12]. It is therefore important that users correct for obvious errors when compiling the four required data to calculate the ε-index (i10, h-index, cm, t). This could include corrections for misattributed articles, start year, or even i10. In some cases, poorly maintained Google Scholar profiles might exclude certain researchers from comparative samples. Regardless, should an assessor have access to potentially more rigorous citation databases (e.g., Scopus), the ε-index can still be readily calculated, although within-sample consistency must be maintained for the ranks to be meaningful. Nonetheless, because the index is relative and scaled, the relative rankings of researchers should be maintained irrespective of the underlying database consulted to derive the input data.”

“The entire approach we present here assumes that each researcher’s Google Scholar profile is accurate, up-to-date, and complete.”

This is a dangerous approach. Real life shows us that profiles are noisy, with errors (some of them on purpose). It is clear that here the important thing is to test the statistical nature of the indicator. However, the sensitive of the indicator to the nature of the database in real conditions can add robustness to this proposal.

RESPONSE #12: Please see the previous response (Response #11) and the new text clarifying this point.

Why authors selected these specific disciplines? Why the number of researchers is equal? The demography of these disciplines is not equal.

RESPONSE #13: Please see Response #1 to Reviewer #1.

I understand the research design and the underlying reasons. However, again, the real conditions of the database should be acknowledged, and decisions should be strongly justified. With the results obtained I cannot be sure if the indicator would be useful for other disciplines and, then, generalizing the strengths of the indicator.

RESPONSE #14: Please see Responses #11 and #15. 

“we did not intend for sampling to be a definitive comment about the performance of particular researchers, nor did we mean for each sample to represent an entire discipline”

While I understand this point, this is important, as authors are trying to operate an indicator with a particular dataset. Biases of this dataset can be inherited in the conclusions achieved. If the sample does not represent the entire discipline, then how I can infer its usefulness to this discipline?

RESPONSE #15: We reiterate that the ε-index is not designed to encompass a discipline; rather, it is designed to compare a sample of individual researchers and compare their relative performance to each other. That someone not included in the sample has a higher h-index or m-quotient (or some other metric) is irrelevant. Because they are not included in the sample, they are not being compared. This is a severe limitation of other metrics because no one can objectively judge what a ‘good’ or ‘bad’ h-index is in any discipline.

“peer-reviewed article published in a recognized scientific journal”

What a recognized scientific journal is for authors in the context of google scholar profiles?

RESPONSE #16: The eight co-authors from eight separate disciplines had no issues in distinguishing peer-reviewed journals from other types of output. We suspect that in extremely rare occasions when a compiler is unfamiliar with the journals in a particular discipline that questions might arise; however, there are several publicly (e.g., Journals Directory; Wikipedia) and subscription-based (e.g., Web of Knowledge; Scopus) databases where one can verify if an article is published in a peer-reviewed journal.

“For the designation of Y1, we excluded any reports, chapters, books, theses or other forms of publication that preceded the year of the first peer-reviewed article; however, we included citations from the former sources in the researcher’s i10, h, and cm.”

I disagree with this procedure. I do not see justifiable to exclude book chapters as document, and later include their citations, it can introduce citation biases. Please justify this decision.

RESPONSE #17: In nearly all cases, reports, chapters, books, theses, etc. listed prior to the first peer-reviewed did not contribute to any of the metrics required for the calculation of ε. None of the researchers’ most highly cited item (cm), h-index, or even i10 were influenced by anything appearing before the first peer-reviewed paper. Even in cases where this might occur (e.g., most likely with the i10), it makes little difference to the relative position of the researcher.

Given that earlier entries are often missing details or have incorrect attribution to the researcher in question (e.g., in Google Scholar), the only way to standardize career length was to establish a rule such as this to identify the start of one’s career. We concede that in disciplines where peer-reviewed articles are the minority of a researcher’s output (e.g., many of the humanities), this type of threshold might be a disadvantage. However, in such cases it is simple to shift the beginning of a researcher’s publication career to a different rule (e.g., first book chapter). Because the number of publication years can be easily manipulated to account for career gaps, it can also easily be manipulated to take into account any aspects of debatable start year discussed above.

All figures performed are of excellent quality and are very informative. The data segregation according to gender is so interesting and adds new debates and discussions. I congratulate authors for this effort in data visualization.

RESPONSE #18: Thank you. No response required.

References cited in this response

1. Phelan TJ. A compendium of issues for citation analysis. Scientometrics. 1999;45(1):117-36. doi: 10.1007/BF02458472.

2. Wildgaard L. An overview of author-level indicators of research performance. In: Glänzel W, Moed HF, Schmoch U, Thelwall M, editors. Springer Handbook of Science and Technology Indicators. Cham: Springer International Publishing; 2019. p. 361-96.

3. Schubert A, Schubert G. All along the h-index-related literature: a guided tour. In: Glänzel W, Moed HF, Schmoch U, Thelwall M, editors. Springer Handbook of Science and Technology Indicators. Cham: Springer International Publishing; 2019. p. 301-34.

4. Bornmann L, Mutz R, Daniel H-D. Are there better indices for evaluation purposes than the h index? A comparison of nine different variants of the h index using data from biomedicine. J Am Soc Inf Sci Tec. 2008;59(5):830-7. doi: 10.1002/asi.20806.

5. Bornmann L, Mutz R, Hug SE, Daniel H-D. A multilevel meta-analysis of studies reporting correlations between the h index and 37 different h index variants. J Informetr. 2011;5(3):346-59. doi: 10.1016/j.joi.2011.01.006.

6. Egghe L. The Hirsch index and related impact measures. Annual Review of Information Science and Technology. 2010;44(1):65-114. doi: 10.1002/aris.2010.1440440109.

7. Waltman L, van Eck NJ. The inconsistency of the h-index. J Am Soc Inf Sci Tec. 2012;63(2):406-15. doi: 10.1002/asi.21678.

8. Fenner T, Harris M, Levene M, Bar-Ilan J. A novel bibliometric index with a simple geometric interpretation. PLoS One. 2018;13(7):e0200098. doi: 10.1371/journal.pone.0200098.

9. Schmoch U, Schubert T, Jansen D, Heidler R, von Görtz R. How to use indicators to measure scientific performance: a balanced approach. Res Eval. 2010;19(1):2-18. doi: 10.3152/095820210X492477.

10. Ràfols I. S&T indicators in the wild: contextualization and participation for responsible metrics. Res Eval. 2019;28(1):7-22. doi: 10.1093/reseval/rvy030.

11. Teixeira da Silva JA, Dobránszki J. Multiple versions of the h-index: cautionary use for formal academic purposes. Scientometrics. 2018;115(2):1107-13. doi: 10.1007/s11192-018-2680-3.

12. Teixeira da Silva JA, Dobránszki J. Rejoinder to “Multiple versions of the h-index: cautionary use for formal academic purposes”. Scientometrics. 2018;115(2):1131-7. doi: 10.1007/s11192-018-2684-z.

---

## [Decision Letter · Decision Letter 1]

25 Aug 2021

A fairer way to compare researchers at any career stage and in any discipline using open-access citation data

PONE-D-21-03358R1

Dear Dr. Bradshaw,

We’re pleased to inform you that your manuscript has been judged scientifically suitable for publication and will be formally accepted for publication once it meets all outstanding technical requirements.

Kind regards,

Sergi Lozano

Academic Editor

PLOS ONE

Additional Editor Comments (optional):

Reviewers' comments:

Reviewer's Responses to Questions

**Comments to the Author**

1. If the authors have adequately addressed your comments raised in a previous round of review and you feel that this manuscript is now acceptable for publication, you may indicate that here to bypass the “Comments to the Author” section, enter your conflict of interest statement in the “Confidential to Editor” section, and submit your "Accept" recommendation.

Reviewer #2: All comments have been addressed

Reviewer #3: All comments have been addressed

2. Is the manuscript technically sound, and do the data support the conclusions?

Reviewer #2: Yes

Reviewer #3: Partly

3. Has the statistical analysis been performed appropriately and rigorously? 

Reviewer #2: Yes

Reviewer #3: Yes

4. Have the authors made all data underlying the findings in their manuscript fully available?

Reviewer #2: Yes

Reviewer #3: Yes

5. Is the manuscript presented in an intelligible fashion and written in standard English?

Reviewer #2: Yes

Reviewer #3: Yes

6. Review Comments to the Author

Reviewer #2: Authors have addressed correctly all my previous doubts and concerns. The revised manuscript has also fixed some minor errors. I believe the manuscript offers new interesting findings to the discipline.

Reviewer #3: I think the authors did a good job addressing most of the comments in the previous round of reviews. Optionally, I suggest them to reconsider their response #4 to Reviewer #1, as I think that evaluating the correlation between their subjective opinions (or those of other experts) of the researchers in their pool and the ranking they obtain from their index would add substantial value to the paper. They clearly went through a lot of effort to assemble a large multidisciplinary team, so in my opinion this is very much low hanging fruit.

7. PLOS authors have the option to publish the peer review history of their article (what does this mean?). If published, this will include your full peer review and any attached files.

Reviewer #2: No

Reviewer #3: No

---

## [Editor Report · Acceptance letter]

31 Aug 2021

PONE-D-21-03358R1 

A fairer way to compare researchers at any career stage and in any discipline using open-access citation data 

Dear Dr. Bradshaw:

I'm pleased to inform you that your manuscript has been deemed suitable for publication in PLOS ONE. Congratulations! Your manuscript is now with our production department. 

Kind regards, 

on behalf of

Dr. Sergi Lozano 

Academic Editor

PLOS ONE